# Comprehensive cancer-oriented biobanking resource of human samples for studies of post-zygotic genetic variation involved in cancer predisposition

Natalia Filipowicz[1]*, Kinga Drężek[1], Monika Horbacz[1], Agata Wojdak[1], Jakub Szymanowski[1,2], Edyta Rychlicka-Buniowska[1], Ulana Juhas[1], Katarzyna Duzowska[1], Tomasz Nowikiewicz[3,4], Wiktoria Stańkowska[1], Katarzyna Chojnowska[1], Maria Andreou[1], Urszula Ławrynowicz[1], Magdalena Wójcik[1], Hanna Davies[5], Ewa Śrutek[4,6], Michał Bieńkowski[7], Katarzyna Milian-Ciesielska[8], Marek Zdrenka[6], Aleksandra Ambicka[9], Marcin Przewoźnik[9], Agnieszka Harazin-Lechowska[9], Agnieszka Adamczyk[9], Jacek Kowalski[7], Dariusz Bała[4,10], Dorian Wiśniewski[10], Karol Tkaczyński[10], Krzysztof Kamecki[11], Marta Drzewiecka[3], Paweł Wroński[11], Jerzy Siekiera[11], Izabela Ratnicka[12], Jerzy Jankau[12], Karol Wierzba[13], Jarosław Skokowski[14,15], Karol Połom[14], Mikołaj Przydacz[16], Łukasz Bełch[16], Piotr Chłosta[16], Marcin Matuszewski[17], Krzysztof Okoń[8], Olga Rostkowska[18], Andrzej Hellmann[18], Karol Sasim[19], Piotr Remiszewski[18], Marek Sierżęga[20], Stanisław Hać[18], Jarosław Kobiela[18], Łukasz Kaska[18], Michał Jankowski[4,10], Diana Hodorowicz-Zaniewska[20], Janusz Jaszczyński[21], Wojciech Zegarski[4,10], Wojciech Makarewicz[14,22], Rafał Pęksa[7], Joanna Szpor[8], Janusz Ryś[9], Łukasz Szylberg[6,23], Arkadiusz Piotrowski[1,24], Jan P. Dumanski[1,5,24]*

1 3P-Medicine Laboratory, Medical University of Gdańsk, Gdańsk, Poland, 2 Bioenit Jakub Szymanowski, Gdańsk, Poland, 3 Department of Breast Cancer and Reconstructive Surgery, Oncology Center—Prof. Franciszek Łukaszczyk Memorial Hospital, Bydgoszcz, Poland, 4 Surgical Oncology, Ludwik Rydygier's Collegium Medicum, Bydgoszcz, Nicolaus Copernicus University, Toruń, Poland, 5 Department of Immunology, Genetics and Pathology and Science for Life Laboratory, Uppsala University, Uppsala, Sweden, 6 Department of Tumor Pathology and Pathomorphology, Oncology Center—Prof Franciszek Łukaszczyk Memorial Hospital, Bydgoszcz, Poland, 7 Department of Pathomorphology, Medical University of Gdańsk, Gdańsk, Poland, 8 Department of Pathomorphology, Jagiellonian University Medical College, Kraków, Poland, 9 Department of Tumor Pathology, Maria Skłodowska-Curie National Research Institute of Oncology, Kraków, Poland, 10 Department of Surgical Oncology, Oncology Center—Prof. Franciszek Łukaszczyk Memorial Hospital, Bydgoszcz, Poland, 11 Department of Urology, Oncology Center—Prof. Franciszek Łukaszczyk Memorial Hospital, Bydgoszcz, Poland, 12 Department of Plastic Surgery, Medical University of Gdańsk, Gdańsk, Poland, 13 Department of Internal Medicine, Connective Tissue Diseases and Geriatrics, Medical University of Gdańsk, Gdańsk, Poland, 14 Department of Surgical Oncology, Medical University of Gdańsk, Gdańsk, Poland, 15 Department of Medical Laboratory Diagnostics-Biobank, Medical University of Gdańsk, Gdańsk, Poland, 16 Department of Urology, Jagiellonian University Medical College, Kraków, Poland, 17 Department and Clinic of Urology, Medical University of Gdańsk, Gdańsk, Poland, 18 Department of General, Endocrine and Transplant Surgery, Medical University of Gdańsk, Gdańsk, Poland, 19 Clinic of Urology and Oncological Urology, Specialist Hospital of Kościerzyna, Kościerzyna, Poland, 20 Department of General, Oncological, and Gastrointestinal Surgery, Jagiellonian University Medical College, Kraków, Poland, 21 Department of Urology, Maria Skłodowska-Curie National Research Institute of Oncology, Kraków, Poland, 22 Clinic of General and Oncological Surgery, Specialist Hospital of Kościerzyna, Kościerzyna, Poland, 23 Department of Clinical Pathomorphology, Collegium Medicum in Bydgoszcz, Nicolaus Copernicus University, Toruń, Poland, 24 Department of Biology and Pharmaceutical Botany, Medical University of Gdańsk, Gdańsk, Poland

* natalia.filipowicz@gumed.edu.pl (NF); jan.dumanski@igp.uu.se (JPD)



**Data Availability Statement:** We are not able to provide the access to our internal database MABData 2, as this is only available locally for the

authorized researchers of our unit. However, we have prepared a minimal anonymized dataset in excel file with all the donors and clinical/medical data that were included in the paper. We provide this information as a supporting information file (S1 Table).

**Funding:** This study was sponsored by the Foundation for Polish Science (FNP) under the International Research Agendas Program to J.P.D. and A.P., co-financed by the European Union under the European Regional Development Fund. Our biobank also received financing via the "Excellence Initiative - Research University" program from Medical University of Gdansk. This project obtained further partial funding from The Swedish Cancer Society and Swedish Medical Research Council to J.P.D. The funders had no role in study design, data collection and analysis, decision to publish, or preparation of the manuscript.

**Competing interests:** J.P.D. is cofounder and shareholder in Cray Innovation AB. The remaining authors have declared that no competing interests exist.

# Abstract

The progress in translational cancer research relies on access to well-characterized samples from a representative number of patients and controls. The rationale behind our biobanking are explorations of post-zygotic pathogenic gene variants, especially in non-tumoral tissue, which might predispose to cancers. The targeted diagnoses are carcinomas of the breast (via mastectomy or breast conserving surgery), colon and rectum, prostate, and urinary bladder (via cystectomy or transurethral resection), exocrine pancreatic carcinoma as well as metastases of colorectal cancer to the liver. The choice was based on the high incidence of these cancers and/or frequent fatal outcome. We also collect age-matched normal controls. Our still ongoing collection originates from five clinical centers and after nearly 2-year cooperation reached 1711 patients and controls, yielding a total of 23226 independent samples, with an average of 74 donors and 1010 samples collected per month. The predominant diagnosis is breast carcinoma, with 933 donors, followed by colorectal carcinoma (383 donors), prostate carcinoma (221 donors), bladder carcinoma (81 donors), exocrine pancreatic carcinoma (15 donors) and metachronous colorectal cancer metastases to liver (14 donors). Forty percent of the total sample count originates from macroscopically healthy cancer-neighboring tissue, while contribution from tumors is 12%, which adds to the uniqueness of our collection for cancer predisposition studies. Moreover, we developed two program packages, enabling registration of patients, clinical data and samples at the participating hospitals as well as the central system of sample/data management at coordinating center. The approach used by us may serve as a model for dispersed biobanking from multiple satellite hospitals. Our biobanking resource ought to stimulate research into genetic mechanisms underlying the development of common cancers. It will allow all available "-omics" approaches on DNA-, RNA-, protein- and tissue levels to be applied. The collected samples can be made available to other research groups.

## Introduction

One of the prerequisites for translational research is availability of well-characterized samples of different types from patients suffering from various diseases. Another requirement is the access to comprehensive and long-term follow-up clinical records for patients, which is important for the correlations between molecular findings and medical parameters. The third condition is a broad participation of patients treated at hospitals as well as control subjects, via their donation of samples to research projects. When these conditions are met, progress can be made towards 3P Medicine, i.e. preventive, personalized and precision.

Cancer is generally defined as a genetic disease, but the frequency of germline cancer-predisposing mutations vary considerably between different tumors and these inherited mutations are responsible for less than 10% of all cancers [1–3]. The remaining >90% of cancers arise as a result of mutations acquired during lifetime in normal somatic cells and the bulk of all cancers occurs late in life. Studies of cancer genomes have contributed to numerous discoveries of mutations that drive cancer growth. However, a fully developed tumor is often clonally heterogeneous and represents a late stage of the disease. This might restrict description of mutations that are occurring very early and initiate tumor development.

Post-zygotic (or somatic) mutations (PZM) in histologically normal human cells from various organs that develop cancer have increasingly been suggested over the past decade as a major source of cancer driving mutations, but this field is still poorly explored [4]. Examples can be given for breast cancer [5–7], normal skin and other normal tissues [8,9], colon cancer [10], urinary bladder cancer [11], aberrant clonal expansions in peripheral blood of healthy subjects (also known as clonal hematopoiesis of indeterminate potential) [12–14] and esophageal cancer [15]. In the latter study, the prevalence of cancer driving mutations was higher in normal epithelium than in esophageal cancers. Furthermore, our biobanking project has been influenced by our interest in analysis of mosaic Loss of chromosome Y (LOY) in blood. It has been noted for over 50 years that chromosome Y is frequently lost in the leukocytes of aging men [16,17]. Recent epidemiological investigations show that LOY in leukocytes, representing lack of nearly 2% of the haploid nuclear genome, is associated with earlier mortality and morbidity from many diseases in men, including multiple common forms of cancer [13,18,19]. Moreover, LOY is the most common post-zygotic (somatic) mutation from analyses of bulk DNA and single-cells from peripheral blood [20].

Thus, comprehensive cancer-oriented biobanking requires sampling of not only tumor tissues but also normal fragments from the affected organ and other control tissue(s) that is not directly involved in the disease process. We use the term "uninvolved margin (UM) or tissue" that refers to histologically controlled non-tumorous tissue, which is located at various distances from the site of primary tumor [5,6]. Collection of blood and plasma (liquid biopsy) is also crucial for future genetic and proteomic analyses. We report here the results of biobanking activities for five common cancer diagnoses that have been ongoing at five major clinical cancer centers in four cities of Poland for a period of more than 23 months.

## Materials and methods

### Diagnoses, logistics and collection protocols

We selected five diagnoses, i.e. breast-, colorectal-, prostate-, bladder- and exocrine pancreas carcinomas, as well as metachronic metastases of colorectal cancer to liver. The choice was based on the high incidence of these diagnoses and/or often fatal outcome of the disease. We established collaboration with five clinical centers in Poland: Oncology Center in Bydgoszcz; National Institute of Oncology in Cracow; University Clinical Centre in Gdansk, University Hospital in Cracow and Specialist Hospital in Koscierzyna. The Pathology Departments at each of these centers were equipped with small -80˚C freezers (80-liter volume, model ULTF 80, Arctico), used for temporary storage of samples prior to shipment to 3P-Medicine Laboratory in Gdansk. The freezers were accompanied by a laptop computer and rack reader for 2D Data-Matrix coded tubes (2 ml, Micronic), used for registration of samples with help of newly developed "MABData1" biobanking software (see below). Dispatch of samples on dry-ice and the corresponding documents from each hospital to 3P-Medicine Laboratory in Gdansk takes place approximately quarterly according to a well-defined procedure.

Figs 1 and 2 show types and number of independent samples collected for two common diagnoses such as breast- and prostate cancer. Breast cancer samples were obtained during mastectomy or breast conserving therapy (BCT) surgeries. Fig 3 show similar outlines of sample collection from patients operated for colorectal cancer (resection of primary tumor and metachronic metastasis of colorectal cancer to liver), bladder cancer (treated by either cystectomy or Trans-Urethral Resection of Bladder Tumor [TURBT]) and exocrine pancreas cancer, respectively. For each diagnosis, the compulsory set of samples include: 1–2 primary tumor fragments (PT); 1–12 specimens of uninvolved margin (UM) composed of macroscopically normal tissue collected at various distances from PT; 1–4 samples of whole blood (WB) (1.5 ml

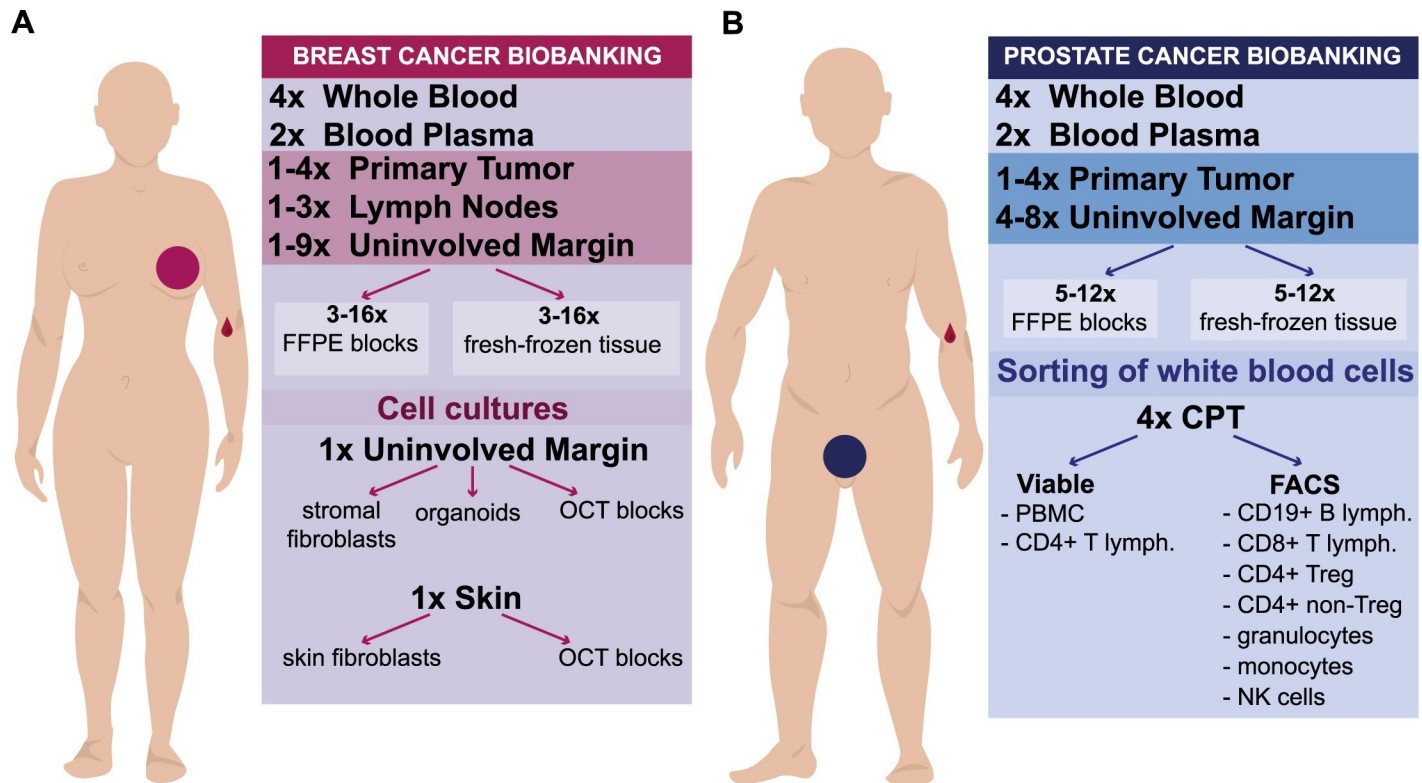

**Fig 1. A summary of samples collected for two common cancer diagnoses.** (A) Collection for breast carcinoma patients. (B) Collection for prostate carcinoma patients. FFPE, Formalin-Fixed Paraffin-Embedded blocks; OCT, Optimal Cutting Temperature compound for fresh-frozen tissue; PBMC, Peripheral Blood Mononuclear Cells; CPT, Cell Preparation Tube with Sodium Heparin (BD Bioscience) for separation of granulocyte- and PBMC-fraction of white blood cells; FACS, Fluorescent Activated Cell Sorting; lymph., lymphocytes; Treg, T-regulatory lymphocytes; NK, Natural Killer cells.

each) as the normal reference tissue; and 1–2 samples blood plasma (BP) (1–1.5 ml each) for future proteomic studies. Whenever available, for breast and colorectal carcinoma, local metastases to lymph node(s) (LN) were also collected, but only when they were clearly identifiable and large enough on gross examination. The volumes of samples from solid tissues ranged between 0.005 cm$^3$ to 1 cm$^3$. The tissues were collected according to the well-defined protocols. After macro-sectioning of the resected organ, small tissue fragments were selected and excised for biobanking. Subsequently, each fragment was cut in half: one portion was placed into a cryovial and fresh-frozen at -80˚C, while the other one was fixed in formalin, embedded in paraffin and underwent standard processing, sectioning and H&E staining (FFPE). The latter FFPE tissue sectioning was done along the cut surface closest to the fresh-frozen biobanked piece of tissue, so that the FFPE section is as much as possible representative for the tissue in the frozen specimen. Therefore, despite the degree of uncertainty/discrepancy in the macroscopic assessment in some situations (i.e. for prostatic adenocarcinoma, multifocal breast cancer, pancreatic adenocarcinoma with coexistent chronic pancreatitis, tumors after neoadjuvant therapy), every single biobanked tissue fragment (both tumor and normal) has its matching FFPE tissue that undergoes pathological verification of the actual tumor content.

The dispersed nature and scale of our biobanking required development of unified sample collection protocols, using well-defined clinical criteria for patient inclusion (Table 1). These protocols were developed in close collaboration between the molecularly-oriented team, surgeons involved in patient recruitment and treatment and pathologists collecting samples. In planning for material collection, we relied on our previous experience from studies of breast

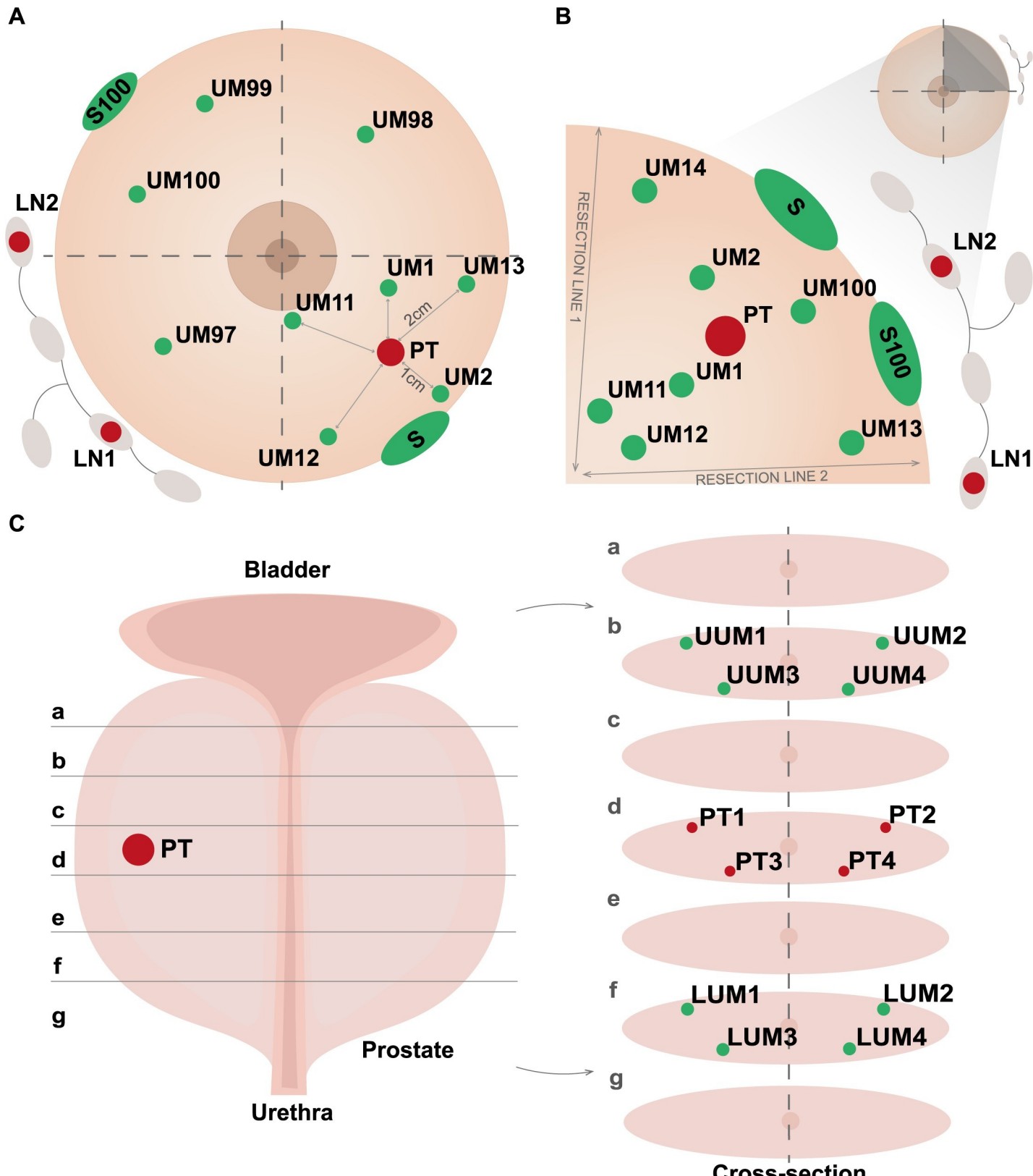

**Fig 2. An illustration of sample collection protocols for breast- and prostate cancer.** (A) Procedure for breast carcinoma samples treated with mastectomy. (B) Procedure for breast carcinoma patients treated with Breast Conservative Therapy (BCT). The distances in centimeters between samples of primary tumor and normal

tissue are illustrated in panel A with solid lines. (C) Protocol for prostatectomy with detailed scheme of sample collection in different cross-sections (a–g). Abbreviations: UM, uninvolved margin composed of macroscopically normal tissue; PT, primary tumor; S, skin; LUM, lower uninvolved margin; UUM, upper uninvolved margin; LN, regional lymph node. Detailed description of particular fragments for the protocols is given in the Materials and Method section.

cancer [5,6,21]. In order to assure good quality of RNA/DNA extracted from the collected material, the general condition for all collected samples was a standardized time of usually 60–75 minutes (and maximum 2 hours) between tumor/organ resection and the moment of -80-degrees freezing for specimens dissected by the pathologist. It did not apply for the material used for Fluorescent Activated Cells Sorting (FACS) of leukocytes in the context of Loss Of chromosome Y (LOY) in male colorectal- and prostate cancer study as well as for material used for establishment of primary cell cultures (see below).

Tissue fragments dissected from mastectomy (Fig 2A) specimen include 1–4 PTs (depending on the tumor size and multifocality) and 9 UMs. Samples UM1 and UM2 located ~1 cm from PT (and presumably in the same lobe of the breast) towards the nipple and outer part, respectively. UMs 11–13 located ~2 cm from PT in the same quadrant, UMs 97 and 98 in two adjacent quadrants, UM99 in opposite quadrant, and S (skin) sample close to resection margin. BCT protocol (Fig 2B) involves collection of 1–2 PTs (depending on the tumor size), 7 UMs and 1 S. UMs 11–14 are collected in the vicinity of both resection lines; two of them (UMs 11 and 12) are closer to the nipple, while the other two (UMs 13 and 14) are further away from the center of the breast. Sample UM1 is taken between the nipple and PT, UM2 between PT and outer part of the breast. All the latter UMs are presumed to be located within the same quadrant and possibly the same lobe, in which the tumor is localized. In both breast cancer procedures, local metastases (LN) are also collected, when possible. Moreover, for both breast cancer surgical procedures, S100 and UM100 are dissected by surgeons directly at the operation theatre to a tube with sterile medium with antibiotics and sent to Gdansk within 24 hours to establish organoids and primary cell cultures (see below).

Due to the usually multifocal growth of prostate adenocarcinoma (constituting up to 95% of all prostate cancers) within this organ and frequent problems to macroscopically assess its exact location, the whole gland is sliced from base to the apex (Fig 2C). Four samples are taken with the punch in the slice that likely include the tumor: PT1 peripheral, right lobe; PT2 peripheral, left lobe; PT3 periurethral, right lobe; PT4 periurethral, left lobe. Analogically, four fragments of potentially unaffected tissue are collected in the slice located towards the base (UUM1–4) and apex (LUM1–4). A total of twelve tissue fragments are biobanked, each followed by the pathological report on the actual tumor content in the matching FFPE sample.

For colorectal cancer, up to 4 PTs (labeled PT1 and PT2 in case of multifocality, or A and B when two fragments are resected) and 8 UMs of mucosa (without muscular and serosa layer) were preserved: UMs 1–3 collected 1 cm away from the margin of PT, UMs 11–13 with 2 cm distance from the tumor, UM98 and 99 at least 5 cm from PT, with UM99 as the most distant one, located as far as possible from PT (Fig 3A). For metachronic metastasis (MT) of colorectal cancer to liver two fragments MT1 and MT11 (in the center and margin of the metastatic tumor respectively) and maximum two UM (close—UM1 and distant—UM99) are collected (Fig 3B). A similar pattern of samples is applicable for radical cystectomy (Fig 3C), while a unique protocol for TURBT involves one PT and two UM samples (Fig 3D). Collection for exocrine pancreatic cancer involves one PT and 1–3 UMs (depending on the type of resection and size of the material that is available) located about 1 cm from the margin of PT, while the UM99 is being the most remote fragment from PT (Fig 3E and 3F).

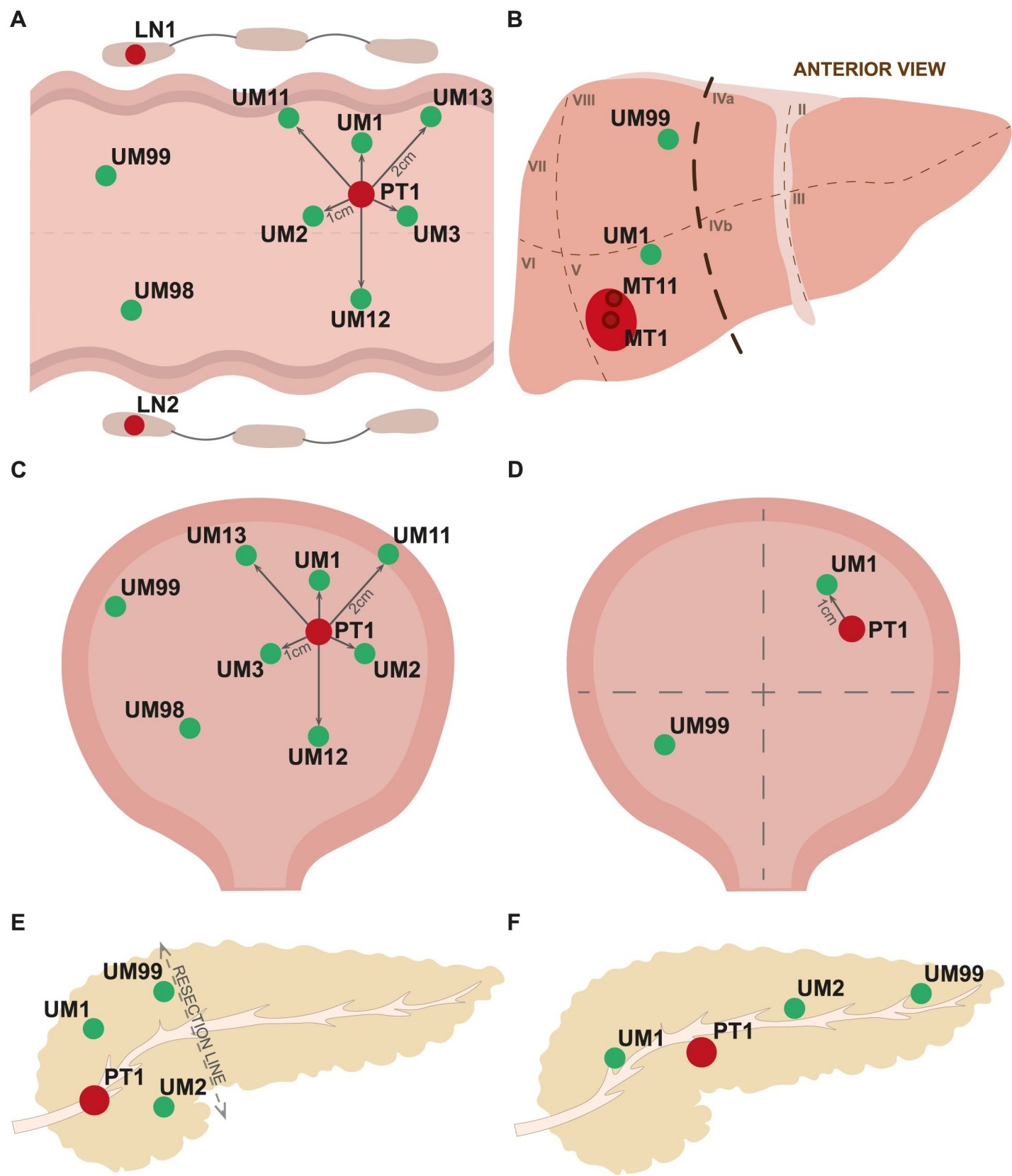

**Fig 3. An illustration of sample collection protocols for colorectal carcinoma and metastases of colorectal cancer to liver, urinary bladder- and exocrine pancreas carcinomas.** (A) Scheme of sample collection for colorectal carcinoma. (B) Protocol of samples collection for metastases of colorectal cancer to liver. (C) Protocol of sample collection for urinary bladder after cystectomy. (D) Collection of samples for transurethral resection of tumor (TURBT). (E) Scheme of sample collection for the surgical removal of pancreas head. (F) Scheme of sample collection for the total pancreatomy. Primary tumors in all panels are drawn in red and samples of normal tissues in green. Abbreviations: UM, uninvolved margin composed of macroscopically normal tissue; PT, primary tumor; LN, regional lymph node; the lines show distances in centimeters from primary tumor.

## Cell cultures for breast cancer- and sorted leukocytes for Loss Of chromosome Y (LOY) project

In addition to the standard deep-frozen samples for biobanking, we also gathered additional unique material demanding dedicated procedures. Primary cultures from skin and uninvolved non-tumorous glandular tissue fragment of breast cancer patients (Fig 2A and 2B, S100 and UM100) yielding skin and stromal fibroblasts as well as organoids were initiated for a subset of cases using in house designed primary cell-culture protocols. For the same subset of patients, OCT blocks from skin and uninvolved margin were also prepared as a possible material for future spatial transcriptomics analysis.

In order to further study the association between LOY in blood and prostate- as well as male colorectal cancer, a total volume of 36 ml of peripheral blood was used for leukocyte sorting using FACS, as described previously [20]. Blood processing involved viable freezing of 2–4 million of peripheral blood mononuclear cells (PBMCs) and 0.1–4 million of CD4[+] cells for single cell analysis. The sorted fractions included: CD19[+] B cells, CD8[+] T cells, CD4[+] T-regulatory (Treg) and CD4[+] non-Treg cells, granulocytes, monocytes and natural killer (NK) cells for further DNA and RNA analysis. We also collected samples from whole blood and buccal swabs for males without cancer or Alzheimer disease diagnosis that were age-matched to the above-mentioned cohorts of prostate- and male colorectal cancer patients. These subjects will serve as controls and their blood leukocytes were sorted and preserved according to the above-mentioned scheme.

## Development of dedicated IT solutions

The computerization and semi-automation of the sample collection was implemented from the very beginning at each collection site. This facilitated the process of preserving a very

**Table 1. Inclusion and exclusion criteria for each diagnosis.**

| Diagnosis | Inclusion criteria |
|---|---|
| Breast cancer | BCT with/without neoadjuvant therapy<br>Unilateral or bilateral mastectomy with/without neoadjuvant therapy |
| Colorectal cancer | Resection of uni- or multifocal primary tumor with/without neoadjuvant therapy |
| Liver metastasis | Resection of uni- or multifocal metachronous tumor with/without perioperative therapy |
| Prostate cancer | Prostatectomy with/without neoadjuvant therapy |
| Bladder cancer | TURBT with/without neoadjuvant therapy<br>Radical cystectomy with/without neoadjuvant therapy |
| Pancreatic cancer | Uni- or multifocal *Adenocarcinoma* (exocrine cancer) *<br>Pancreaticoduodenectomy ** without neoadjuvant therapy<br>Total pancreatectomy without neoadjuvant therapy |
| Control group | Age ≥ 65 y.o. without oncological and Alzheimer Disease in clinical history |

BCT–Breast Conservative Therapy; metachronic tumor stands for the secondary tumor that arose more than 6 months after diagnosis of first malignancy; TURBT–Transurethral Resection of Bladder Tumor, *—exclusion criteria: Preoperative neoadjuvant therapy, Endocrine cancer; **—Pancreaticoduodenectomy (Whipple procedure)–operation performed to remove the cancerous head of the pancreas.

high number of donors, reaching 100 per month without the risk of mix-up of samples. Each Pathology Department at the partner hospital was provided with a PC and a dedicated MABData1 software designed for that purpose. This program package has a simple, user-friendly web browser-based interface enabling registration of patients/samples with a set of clinical data (excluding personal information for safety reasons) and automated registration of tubes containing unique 2D codes. It also allows introduction of medical follow-up information at a later stage. All data is being synchronized every 5 minutes with MABServer in Gdansk using safe Advanced Encryption Standard (AES). The main goal of developing dedicated software for donor and sample registration at satellite hospitals was to introduce a user-friendly system that allows fast data synchronization with the central server, independence from network access and proper function also in an unstable internet connection environment. MABServer software is a proxy system (located at the 3P-Medicine Laboratory at the Medical University of Gdansk) that holds data from all hospitals for further use. Moreover, this system was developed also for safety reasons; in case of any MABData1 computer failure all data is kept at MABServer, which is included in the central backup schedule at the 3P-Medicine Laboratory. The management of samples and data locally is being assisted with MABData2 system, allowing donor and sample pseudonymization, registration of original samples from hospitals and numerous types of derivative specimens, together with many additional parameters, adding extra attachments, simple and advanced searching and data exporting, as well as inventory of samples. MABData2 is a complete, stand-alone biobank management system that supports not only current biobanking project, but is also the main software solution, connecting data from other projects and collections for further cross-searches. The documentation received on paper is pseudonymized, scanned and uploaded to the MABData2 system, using implementation of Optical Character Recognition (OCR) algorithms inserted into the database.

## Bioethical approval

All procedures for sample collection were approved by the Independent Bioethics Committee for Research at the Medical University of Gdansk (approval number NKBBN/564/2018 with multiple amendments). This approval is valid for collection of samples at multiple collaborating hospitals. Since our collection of samples is still ongoing, our initial ethical approval provides us with the possibility to extend the scope of biobanking for additional diagnoses, after an amendment of the application. Written informed consent was obtained from all the patients prior to surgery. All procedures were performed in accordance with the relevant national and international laws and guidelines as well as in compliance with European Union General Data Protection Regulation (EU GDPR).

## Other laboratory procedures

DNA extraction from frozen solid tissues was performed using standard phenol/chloroform method with several in-house modifications (depending on the tissue type) and lysis buffer with SDS (0.5% SDS, 60mM Tris pH 7.9, 5 mM EDTA) or sarcosine and SDS (2% sarcosine, 0.5% SDS, 50mM Tris pH 7.9, 10 mM EDTA); DNA from whole blood was acquired using QIAamp DNA Blood Midi Kit (Qiagen) or QuickGene DNA whole blood kit S (Kurabo) with QuickGene-Mini 480 instrument (Kurabo), DNA from sorted cells was extracted either using sarcosine lysis buffer (1% sarcosine, 10 mM Tris-HCl, 50mM NaCl, 10 mM EDTA) followed by ethanol precipitation (<500 000 cells) or with QIAamp DNA Mini Kit (Qiagen) (>500 000 cells). All laboratory processes using our samples were carried out according to standard protocols including: sorting of leukocytes on FACS machines, establishing primary cultures,

extraction of DNA and RNA for various downstream applications; e.g. droplet digital PCR (ddPCR), construction of NGS libraries for targeted DNA/RNA sequencing.

## Results

### Rationale

The main rationale and aims of our biobanking program are systematic explorations PZMs especially in normal tissues, which is an understudied field of cancer research. We also incorporate material for studies of mosaic LOY in males that might predispose to various diseases, with a particular focus on cancer. The selected diagnoses cover sporadic breast-, colorectal-, prostate-, bladder- and pancreatic carcinomas and represent consecutively collected patients, affected by common and often fatal diseases. The setup of our biobank should allow a wide range of "omics"- and other methods to be applied in studies of the collected clinical material.

### General statistics

The first patient was registered in the database on June 5, 2019 and statistics described here are up to May 12, 2021. Our collection originating from five clinical cancer centers, after nearly 2-years of cooperation, reached in total 1963 donors. However, 1711 of these were collected effectively (i.e. with all tissue types from the collection protocols, described above in Materials and Methods as compulsory set) yielding 23226 independent samples (Fig 4A and 4B). This results with an average of 74 donors and 1010 samples collected per month. Incomplete sampling affected 13% of all donors and was caused by several factors, such as small size of tumor resulting in insufficient amount of tissue material, prolonged surgery time and inability to prepare material by pathologists the same day, as well as unresectability of the tumor. The predominant diagnosis was breast carcinoma (933 donors), followed by colorectal- (383 donors), prostate- (221 donors), urinary bladder- (81 donors), exocrine pancreas carcinomas (15 donors) and metachronous metastases of colorectal cancer to liver (14 donors). We also collected blood and buccal swabs from 64 healthy male control subjects that were age matched for the cohort of males with colorectal- and prostate carcinoma (Table 2). The average age of male oncological patients was 67 years (1 SD ±9, range 33–93 years), while for female oncological donors it was 62 years (1 SD ±13, 24–92 years) and for healthy control subjects 71 years (1 SD ±5, 61–91 years).

Fig 4C–4F shows the distribution of ICD-10 codes within particular diagnoses: C50 (malignant neoplasm of breast), C18—C21 (malignant neoplasm of colon, rectosigmoid junction, rectum, anus and anal canal), C67 (malignant neoplasm of bladder), and C25 (malignant neoplasm of pancreas). Within colorectal diagnoses clear predominance of rectal cancers (C20, 99 cases) is notable, followed by tumors of caecum (C18.0, 57 donors) and sigmoid colon (C18.7, 53 cases). In bladder cancer patients, the affected sites include trigone and lateral wall of the bladder (C67.0 and C67.2 and 34 and 25 donors, respectively). The vast majority of pancreatic tumors are located in the body of the organ (C25.0, 12 patients). In addition to ICD-10 diagnoses, we also gathered other clinical information regarding the donors recruited to the project, which is summarized in Table 3. It covers basic clinical data: imaging results for the patient, type of surgery, dates of initial diagnosis and treatment, blood count, as well as full histopathological report with data for microscopic examination of all resected tissue fragments. Moreover, we collected information from each donor via medical questionnaires covering oncological history in the family, chronic illnesses and smoking habits.

The uniqueness of our collection is illustrated by the number of fragments dissected from macroscopically normal cancer-neighboring margin of tissue (named UM, LUM or UUM) (Figs 1 and 2), which accounts for nearly 40% of the total sample count, while the contribution from

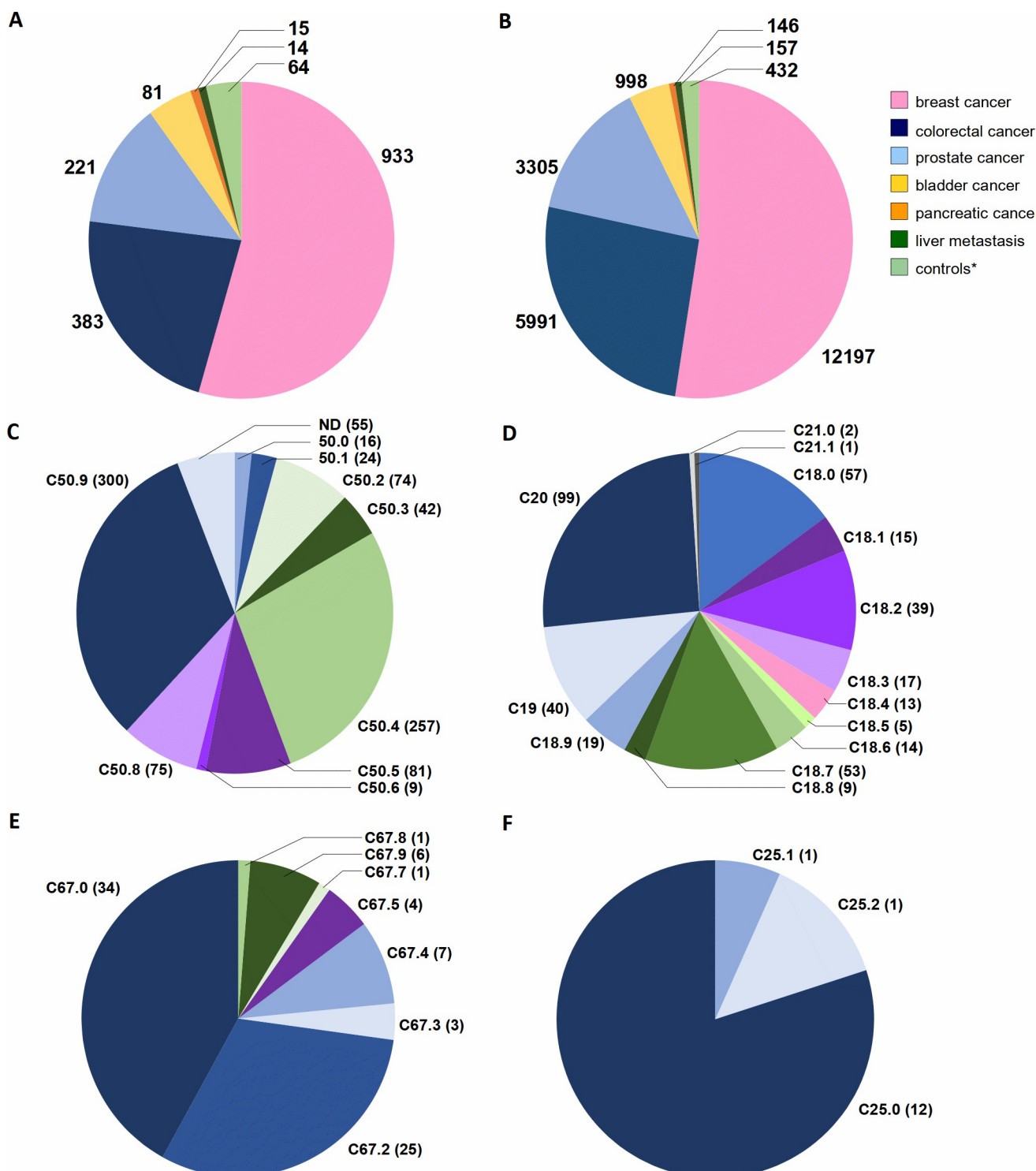

**Fig 4. The statistics of donors and samples collected in five collaborating hospitals; status as of May 12, 2021.** (A) The total number of donors with compulsory set of samples (as described in Materials and Methods-section and shown in Figs 1 and 2). (B) The sum of all samples collected from recruited donors. The numbers for donors and samples are divided for six different cancer diagnoses and controls (*). The control (*) category represents a healthy male cohort ≥ 65-year-old recruited as controls for patients with prostate- and colorectal cancer, used in the Loss of Y Chromosome (LOY) project. (C–F) show distribution of diagnoses according to International Classification of Diseases (ICD-10, World Health Organization) for breast, colorectal, bladder and pancreatic cancer patients, respectively. Abbreviations: C50, Malignant neoplasm of breast; C50.0, Nipple and areola; C50.1, Central portion of breast; C50.2, Upper-inner quadrant of breast; C50.3, Lower-inner quadrant of breast; C50.4, Upper-outer quadrant of breast; C50.5, Lower-outer quadrant of

breast; C50.6, Axillary tail of breast; C50.8, Overlapping lesion of breast; C50.9, Breast, unspecified; C18, Malignant neoplasm of colon; C18.0, Caecum, Ileocaecal valve; C18.1, Appendix; C18.2, Ascending colon; C18.3, Hepatic flexure; C18.4, Transverse colon; C18.5, Splenic flexure; C18.6, Descending colon; C18.7, Sigmoid colon, Sigmoid (flexure); C18.8, Overlapping lesion of colon;C18.9, Colon, unspecified, Large intestine, unspecified; C19, Malignant neoplasm of rectosigmoid junction, including colon with rectum, rectosigmoid colon; C20, Malignant neoplasm of rectum, Including rectal ampulla; C21, Malignant neoplasm of anus and anal canal; C21.0, Anus, unspecified, excluding anal margin and perianal skin; C21.1, Anal canal, Anal sphincter; C67, Malignant neoplasm of bladder; C67.0, Trigone of bladder; C67.2, Lateral wall of bladder; C67.3, Anterior wall of bladder; C67.4, Posterior wall of bladder; C67.5, Bladder neck, Internal urethral orifice; C67.7, Urachus; C67.8, Overlapping lesion of bladder; C67.9, Bladder, unspecified; C25, Malignant neoplasm of pancreas; C25.0, Head of pancreas; C25.1, Body of pancreas; C25.2, Tail of pancreas. ND–not yet defined due to temporary lack of medical documentation.

tumor samples is 12%. The remaining sample represent blood, plasma, skin and sorted leukocytes using FACS. Furthermore, whenever possible we also collected material from local metastases to lymph nodes, for a total of 117 and 80 donors with breast and colorectal cancer, respectively (1% of all samples). The implemented sampling procedures are demanding in terms of time needed for material preparation. The estimated average time devoted by the Pathology Departments is 98 minutes per patient to fulfill the requirements of our protocols (ranging from 60 minutes for liver metastasis resection to 155 minutes for prostatectomy), which include patient registration, blood processing for collection of tubes with peripheral blood and plasma, tissue dissection by pathologists, preparation of FFPE blocks/slides, and histopathological assessment. Thus, the collection from 1711 donors sums up to about 2800 working hours or about 70 working weeks with full time effort. This time represents only part of the entire project, as we do not include time spent on the following tasks: recruitment of patients by surgeons, collecting

**Table 2. A summary of donors and cancer diagnoses included in the collection (status as of May 12, 2021).**

| Diagnosis | Sex distribution | Average age | Average samples (range) | Clinical information |
|---|---|---|---|---|
| Breast cancer (ICD-10 C50) | F– 99% (n = 921) M– 1% (n = 12) | 60 y ± 13 64 y ± 8 | 13 (7–32) | • Mastectomy– 42% (n = 391) • BCT– 53% (n = 494) • Non-specified* – 5% (n = 48) |
| Colorectal cancer (ICD-10 C18 –C21) | F– 45% (n = 174) M –55% (n = 209) | 66 y ± 12 67 y ± 10 | 16 (7–27) | |
| Liver metastasis (colorectal cancer) (ICD-10 C78.7) | M– 86% (n = 11) F– 14% (n = 3) | 65 y ± 11 66 ± 4 | 11 (7–15) | |
| Prostate cancer (ICD-10 C61) | M– 100% (n = 221) | 65 y ± 7 | 15 (7–19) | Gleason score:<br><br>• 3+2–0.5% (n = 1) • 3+3–16% (n = 35) • 3+4–28% (n = 61) • 4+3–28% (n = 61) • 4+4–2% (n = 5) • 4+4+5–0.5% (n = 2) • 4+5–1% (n = 3) • Non-specified* – 24% (n = 53) |
| Bladder cancer (ICD-10 C67) | F– 21% (n = 17) M– 79% (n = 64) | 68 y ± 8 69 y ± 9 | 12 (7–17) | • TURBT– 47% (n = 38) • Cystectomy– 53% (n = 43) |
| Pancreatic cancer (exocrine, ICD-10 C25) | Female– 47% (n = 7) Male– 53% (n = 8) | 68 y ± 6 66 y ± 9 | 10 (8–11) | |
| Control group** | Male– 100% (n = 64) | 71 ± 5 | 7 (6–12) | |

F–female; M–male; *—information not yet available and incorporated in our database; y–years; BCT–Breast Conservative Therapy. ICD codes: C50, Malignant neoplasm of breast; C18, Malignant neoplasm of colon; C19, Malignant neoplasm of rectosigmoid junction; C20, Malignant neoplasm of rectum; C21, Malignant neoplasm of anus and anal canal; C78.7, Secondary malignant neoplasm of liver and intrahepatic bile duct; C61, Malignant neoplasm of prostate; C67, Malignant neoplasm of bladder; C25, Malignant neoplasm of pancreas; TURBT–Transurethral Resection of Bladder Tumor.

**—Healthy male cohort ≥ 65 years old recruited as controls for the male patients with prostate and colorectal cancer for whom the white blood cell fractions were sorted by FACS to study loss of chromosome Y.

**Table 3. Type of data collected from medical records of patients.**

| *Document* | *Information* |
| --- | --- |
| Registration form in the hospital | • Clinical data available from the hospitals: CT, PET, MRI, RTG, USG, colonoscopy, mammography, urography, cystoscopy, scintigraphy;<br>• Date of first diagnosis and treatment;<br>• Type of surgery (radical cystectomy/TURBT, mastectomy/BCT, type of pancreatectomy). |
| Histopathological report | • Histopathological type of cancer;<br>• Microscopic description of tumor/non-tumoral tissue;<br>• Grade of cancer;<br>• pTNM stage;<br>• Gleason score (ICD-10 C61);<br>• ER, PR, HER2 status (ICD-10 C50);<br>• Ki-67 (ICD-10 C50);<br>• Size of resected organ. |
| Patient questionnaire | • Smoking status;<br>• Chronic illnesses;<br>• Family history–oncological treatment, Alzheimer Disease. |
| Complete blood count | • Red Blood Count;<br>• White Blood Count;<br>• Platelets. |

CT–Computer Tomography; PET–Positron Emission Tomography; MRI–Magnetic Resonance Imaging; RTG–radiography; USG–ultrasonography. ICD codes: C50, malignant neoplasm of breast; C61, malignant neoplasm of prostate; TURBT, Transurethral Resection of Bladder Tumor; BCT, Breast Conservative Therapy; ER, Estrogen Receptor status; PR, Progesterone Receptor status; HER2, status of the human epidermal growth factor receptor; Ki-67, marker of proliferation Ki-67.

informed consents and filling of questionnaires, local transport within each hospital and from five partner hospitals to Gdansk, as well as acquisition of material by the 3P-Medicine Laboratory in Gdansk. Furthermore, in the time devoted to development and testing of software (MAB-Data1 and MABData2) we do not count preparation of common protocols for sample collection as well as preparation of formal agreements with the participating hospitals.

Another important aspect of our biobanking approach is the histopathological verification of the tissue fragments that were macroscopically presumed to represent either tumor or non-tumor fragment and were later verified by analysis of FFPE sections. We performed such comparison for prostate cancer and colorectal cancer. As suspected, prostate cancer represents a challenge due to the lack of clear macroscopic demarcation of the tumor. The calculation on the representative cohort of 100 prostate cancer patients showed that out of 400 specimens collected presumably as tumorous tissue, 263 (65.7%) showed the presence of tumor cells. For the similar sample size of 100 colorectal cancer patients, 205 out of 205 (100%) presumed tumor samples were confirmed as containing tumor tissue. It is important to mention that each fragment selected for further molecular examination is verified with the histopathological report.

Analysis of smoking status, which is included in the medical questionnaire (Fig 5), revealed that nearly 50% of female cancer patients declared themselves as non-smokers and 31% as past-smokers. Corresponding numbers of non-smokers among males is clearly lower (28%) and past-smokers is much higher (54%). Similar numbers of females and males declared themselves as present smokers (16% and 14%, respectively), and this is also valid for control subjects (19%).

## Specific diagnoses

The summary of donors and samples for all diagnoses is shown in Table 2. Breast cancer is the most common type of cancer diagnosed world-wide in 2020 [22] (https://www.who.int/news-

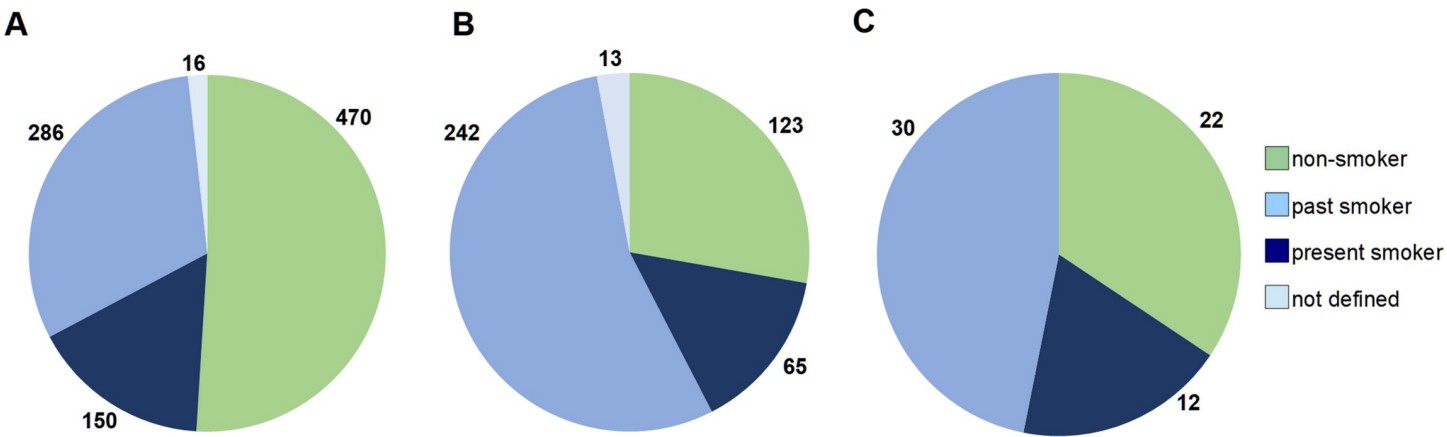

**Fig 5. Smoking status declared by donors in the questionnaire.** (A) Female cohort. (B) Male cohort. (C) Control male group, as described in Materials and Methods.

room/fact-sheets/detail/cancer) and it is also well represented in our collection, accounting for 55% of all donors, with obvious predominance of women and only 1% of males with breast cancer. The average age for females undergoing mastectomy or BCT is 60 years (1 SD ±13), and the corresponding number for males is 64 years (1 SD ±8). Counting all breast surgeries together, 53% are BCTs, and 42% are mastectomies, remaining 5% are not defined due to the temporary lack of medical documentation. Colorectal cancer is the second most common diagnosis in the collection (22% of donors), with the predominance of male patients (55% versus 45% of females), with the average age of initial surgical treatment similar for both sexes (66 versus 67 years). Due to the usually large extent of the resection of the colon, we reached the highest average sample count (n = 16) among all diagnoses. Prostate cancer in our assembly is diagnosed and treated on average at the age of 65 years, at a medium-grade stage: Gleason score 3+4 (27% of patients) and 4+3 (27%), low medium-grade: Gleason score 6 (15%) and rarely at a high-grade, Gleason score ≥8 (3.5%). The collection procedure for prostatectomy is the most complicated of all diagnoses and the average sample count for prostate cancer is the second highest (n = 15). Organ sparing treatment using TURBT, was applied for 47% of all gathered bladder cancer donors, which adds to the uniqueness of our collection. The frequency of bladder cancer among males is much higher than for females (79% versus 21%), with a similar average age of onset for both sexes (69 and 68 years, respectively). Exocrine pancreas cancer, which is frequently unresectable and the most fatal disease in our collection, stands for below 1% of all our donors and has a similar distribution of age and sex.

Well-defined recruitment criteria for volunteers as healthy controls for the LOY project (Table 1) resulted in a homogeneous group of male subjects, with an average age of 71 years and average sample count of seven. The starting material is peripheral blood for preparation of viable PBMCs and CD4+ T-cells and sorting of seven populations of leukocytes using FACS. Additionally, buccal swabs are collected as reference material for non-mesoderm-derived tissue. The same procedure for sorting of leukocytes, as above for controls, was applied for 20 prostate and 28 colorectal cancer patients.

## Quality of derivative samples

The collected samples have already been used for subsequent research activities and a wide variety of downstream applications. Hence, we optimized and implemented several laboratory protocols that include DNA and RNA preparation and aliquoting. We present here the

concentrations and quality of DNA extracted from solid tissue (PT and UMs), blood and the pellets of FACS sorted cells (Fig 6). The amount of tissue used for DNA preparation differs widely for various diagnoses and is lowest for bladder and prostate samples (starting from 2 mg of fresh frozen tissue), where the collected fragments usually do not exceed 8 mm³. However, the quality, integrity (DIN) and spectroscopic purity of DNAs were high for all samples (Fig 6B and 6C) with the exception of A260/230 ratio for DNAs from sorted cell pellets, especially from the subpopulations with low cell count after sorting (below 150 000 cells) (Fig 6C). This quality is, however, still sufficient to run ddPCR analysis.

## Dedicated software for dispersed biobanking

In a typical research project oriented on sample/donor collection, software solutions are built in central models as one system case with a single pipe for data input. In our project, we created a dedicated solution for dispersed biobanking at satellite hospitals. Each partner hospital is actually an independent biobank, equipped with both hardware and software solutions allowing sample/data registration as well as storage, including a possibility of *a posteriori* data verification and supplementation. This developed system and communication methods bring data safety standards that correspond to ISO-270001 requirements. The most important features are included in MABData2 software. This package is not only the tool dedicated for biobanking collection, administration and management, but also allows for customized data search, connecting different types of data and parameters describing samples and donors. Both MABData1 and MABData2 packages are owned by the Medical University of Gdansk, which has full license- and copyrights. Therefore, this in-house developed software allows independence for future development and for any number of users.

## Discussion

One of the major aims of our biobanking effort is to gather a representative collection of samples from multiple common sporadic cancer diagnoses, which would primarily allow large-scale studies of early predisposing PZMs that initiate tumor development in normal cells from

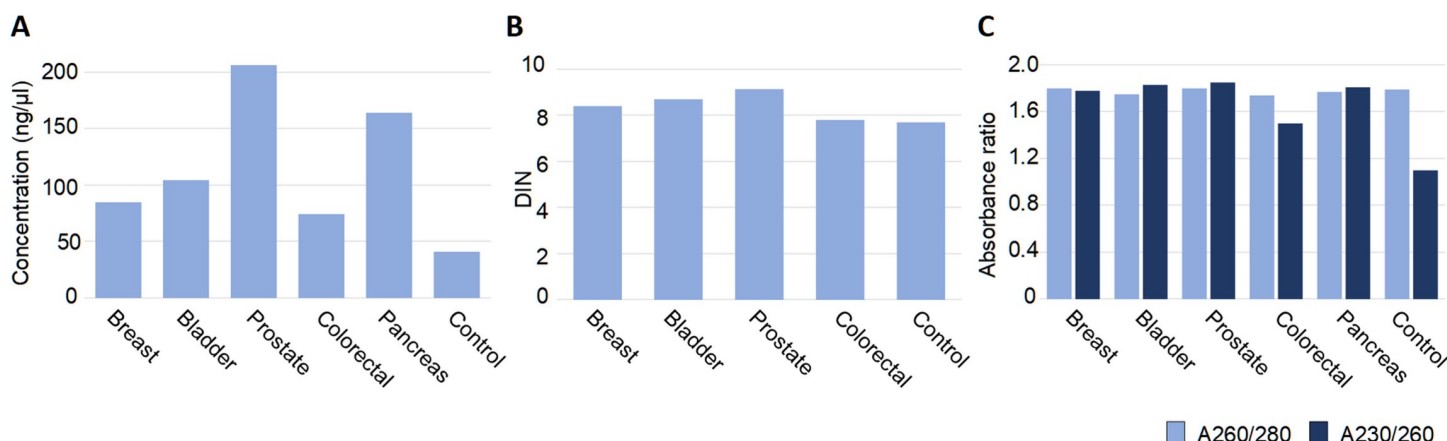

**Fig 6. DNA quality isolated from collected tissues.** (A) Concentration of DNA (ng/μl) obtained from blood and tissues from donors per diagnoses, measured with the fluorometric method (Qubit Fluorometric Quantification and/or Agilent TapeStation System). (B) DNA Integrity Number (DIN) for DNA obtained from blood and tissue in four diagnoses measured by Genomic DNA ScreenTape Analysis (Agilent). (C) DNA quality measured by UV-Vis spectroscopic method for DNA obtained from blood and tissues from donors with five diagnoses. The number of samples for each diagnosis that was used for calculations are: 88 for breast cancer; 202 for colorectal cancer; 3092 for prostate cancer; 189 for bladder cancer; 12 for pancreas cancer; and 62 for controls. The amount of frozen tissue used for DNA extractions range as follows: Breast cancer 15–60 mg; bladder cancer 2–19 mg; prostate cancer 6–43 mg; colorectal cancer 15–33 mg; pancreas cancer 12–21 mg; and controls (sorted leukocytes) 0.05x10⁶ - 1x10⁶ cells.

the affected organ. Consequently, the largest number of samples are derived from histologically verified non-tumorous tissue, which is located at various distances from the site of primary tumor. The design of our biobanking is based on our previous experience from studies of multifocal breast cancer [5,6,21]. Furthermore, availability of material from local metastases to lymph nodes for breast and colorectal cancer as well as distant metastases of colorectal cancer to the liver allows for assessment of tumorous tissue at different stages of disease. Moreover, we collect at least one additional reference tissue that is not related to tumor development; usually blood or skin (Figs 1 and 2). This reference material is important whenever a possibly pathogenic variant is found, in order to confidently exclude that it could represent a germline mutation. The information regarding oncological family history gathered from the patient questionnaire might also be helpful in such cases. Collection of multiple tubes of plasma (liquid biopsy) from donors suffering from cancer and from healthy controls is also crucial for future genetic and proteomic analyses. The highest number of independent samples per single donor is thirty-two (Table 2). Thus, according to the official BBMRI-ERIC directory (https://directory.bbmri-eric.eu/#/), our collection is comprehensive and represents the largest assembly of samples oriented towards studies of early events in cancer development. This collection will allow essentially all available "-omics" approaches on DNA-, RNA-, protein- and tissue levels as well as many other methodological approaches to be applied. Our biobanking project also assumes acquisition of long-term follow-up (3 to 10 years after treatment) for recruited patients, which will be an important added value. As mentioned above, our biobanking effort is still ongoing and we have an opportunity to modify collection protocols and include other cancer diagnoses. The collected biological material and clinical data can be made available for other investigators after a request to both corresponding authors. The letter should outline the aim, number/type of requested samples and methodology of the proposed collaborative project. Such scientific cooperation can be established based on bilateral scientific cooperation agreement.

The number of recruited donors already provides us a good perspective regarding statistics of the four most frequent diagnoses in our study. For the breast carcinoma, a clear predominance (32%) of C50.9 diagnosis (Breast, unspecified) is apparent together with the relative high incidence of ICD 50.8 (Malignant neoplasm of overlapping sites of breast), which suggest unclear location or lack of the record in the patient documentation. This is followed by the carcinomas located in upper outer (ICD 50.4–27%), lower outer (ICD 50.5–9%), and upper inner (ICD 50.2–8%) quadrants of the breast. This trend reflects the distribution observed in other large-scale study [23]. In the case of colorectal cancer, the location of tumor is a crucial factor determining molecular type, disease progression, prognosis, treatment and outcome [24]. In our collection 38% of cases are right-sided colon cancers (RSCC/proximal): ICD-10 from C18.0 to C18.5 and 61% represent left-sided colon cancers (LSCC/distal): C18.6 to C19, C20 (Fig 4D). The distribution is in accordance with a larger study [25]. RSCCs are generally higher in females and have worse overall prognosis—in our study it is comparable for both sexes (21% females vs 19% males), the opposite trend is observed in LSCC (25% for females and 35% in this study). Such a representation and distribution of donors and samples in the collection will enable the examination of molecular heterogeneity, etiology and progression of these cancers.

For breast- and urinary bladder cancer patients, we collected samples derived from two different surgical procedures. Breast cancer operations can be performed using either mastectomy or BCT and the frequency of these two surgeries in our material are 42% and 53%, respectively. The tumors removed via BCT are diagnosed at earlier stage of development, are typically smaller and these patients usually do not show suspicion of multifocal tumor development. The trend in our cohort reflects the worldwide shift in surgical approach

from mastectomy to BCT [26]. Bladder cancer patients are qualified for either cystectomy or sparing therapy using (TURBT) and their respective frequencies of these treatments in our collection are 53% and 47%. Hence, we have an opportunity to molecularly study these two common cancers in patients with different grades of severity of the disease, which represents an added value.

It is well known that males have a higher incidence and mortality from most sex-unspecific cancers, which is largely unexplained by known risk factors [27,28]. This substantial cancer-related sex-disparity appears to be a neglected aspect of cancer research. Our collection of donors for bladder- and colorectal cancer confirms this. Bladder cancer is much more frequent among males than females (79 versus 21), with a similar average age of onset for both sexes (69 and 68 years, respectively) (Table 2). Colorectal cancer also shows male predominance (55% males vs. 45% females), although the difference is less pronounced. Smoking habits among males in Poland might be, at least partially, responsible for these differences. As shown in Fig 5, 47% and 68% oncological female and male patients, respectively, declared themselves as past- or present smokers, and there is a proven correlation between cancer incidence, smoking habits and LOY for men [18,29,30]. Our ongoing effort to obtain sorted subpopulations of leukocytes using FACS from males treated for prostate and colorectal cancer will help to study and possibly provide further support for this hypothesis.

## Supporting information

**S1 Table. Pseudonymized list of patients that were included in the statistics presented in the manuscript with the information on diagnosis, age, sex, ICD10, type of surgery where applicable, number of original samples and the status of smoking declared in the medical questionnaire.**
(XLSX)

## Acknowledgments

We thank all the patients and volunteer controls for acceptance to participate, sample contribution and information provided in the medical questionnaire. We thank Dr. Leszek Kalinowski for use of the temporary office, laboratory and freezer space as well as access to other laboratory facilities. We acknowledge Drs. Darek Kędra, Marco Günthel, and Paweł Olszewski for consultations regarding laboratory and bioinformatic procedures. We also thank physicians and nurses involved in the patient recruitment process, collaborating technicians, diagnosticians and pathologists from: Oncology Center—Prof. Franciszek Łukaszczyk Memorial Hospital in Bydgoszcz (Jowita Nowaczewska, Katarzyna Krzysiak, Anetta Słupicka, Mateusz Matusiak); Maria Skłodowska-Curie National Research Institute of Oncology in Kraków (Justyna Wajda, Dorota Lech, Kaja Majchrzyk); University Clinical Centre in Gdańsk (Wojciech Biernat, Wojciech Połom, Jakub Gondek, Tomasz Cwaliński, Grażyna Stęplewska, Elżbieta Wierszyło, Elżbieta Pietruszka, Grażyna Dombrowska, Paweł Górny, Małgorzata Derwis, Justyna Pietruszewska, Irena Pellowska, Michał Kunc, Aleksandra Korwat, Ewa Miłoszewska, Aleksandra Kaczor, Dawid Foltynowski); University Hospital in Cracow (Weronika Natkaniec, Magdalena Smolik, Małgorzata Molus, Jadwiga Gałek, Hieronim Strojniak, Adrian Głownia, Anna Janas, Iwona Zawadzka, Monika Cała, Joanna Ciężarek, Shymko Anzhela, Izabela Pabisz-Zarębska); and Specialist Hospital in Kościerzyna (Atanasiu Apostolis, Barbara Koenner, Michał Kujach, Renata Knuth). We thank Drs. Daniil Sarkisyan and Eva Tiensuu Janson for critical review of the manuscript.

## Author Contributions

**Conceptualization:** Natalia Filipowicz, Agata Wojdak, Jakub Szymanowski, Jarosław Skokowski, Arkadiusz Piotrowski, Jan P. Dumanski.

**Data curation:** Natalia Filipowicz, Kinga Drężek, Jakub Szymanowski, Jan P. Dumanski.

**Formal analysis:** Natalia Filipowicz, Jakub Szymanowski.

**Funding acquisition:** Arkadiusz Piotrowski, Jan P. Dumanski.

**Investigation:** Natalia Filipowicz, Kinga Drężek, Monika Horbacz, Edyta Rychlicka-Buniowska, Ulana Juhas, Katarzyna Duzowska, Wiktoria Stańkowska, Katarzyna Chojnowska, Maria Andreou, Urszula Ławrynowicz, Magdalena Wójcik.

**Methodology:** Natalia Filipowicz, Monika Horbacz, Agata Wojdak, Edyta Rychlicka-Buniowska, Ulana Juhas, Tomasz Nowikiewicz, Katarzyna Chojnowska, Hanna Davies, Ewa Śrutek, Michał Bieńkowski, Jarosław Skokowski, Krzysztof Okoń, Stanisław Hać, Łukasz Kaska, Michał Jankowski, Diana Hodorowicz-Zaniewska, Rafał Pęksa, Joanna Szpor, Janusz Ryś, Łukasz Szylberg, Arkadiusz Piotrowski, Jan P. Dumanski.

**Project administration:** Natalia Filipowicz, Agata Wojdak, Arkadiusz Piotrowski, Jan P. Dumanski.

**Resources:** Jakub Szymanowski, Tomasz Nowikiewicz, Ewa Śrutek, Michał Bieńkowski, Katarzyna Milian-Ciesielska, Marek Zdrenka, Aleksandra Ambicka, Marcin Przewoźnik, Agnieszka Harazin-Lechowska, Agnieszka Adamczyk, Jacek Kowalski, Dariusz Bała, Dorian Wiśniewski, Karol Tkaczyński, Krzysztof Kamecki, Marta Drzewiecka, Paweł Wroński, Jerzy Siekiera, Izabela Ratnicka, Jerzy Jankau, Karol Wierzba, Jarosław Skokowski, Karol Połom, Mikołaj Przydacz, Łukasz Bełch, Piotr Chłosta, Marcin Matuszewski, Krzysztof Okoń, Olga Rostkowska, Andrzej Hellmann, Karol Sasim, Piotr Remiszewski, Marek Sierżęga, Stanisław Hać, Jarosław Kobiela, Łukasz Kaska, Michał Jankowski, Diana Hodorowicz-Zaniewska, Janusz Jaszczyński, Wojciech Zegarski, Wojciech Makarewicz, Rafał Pęksa, Joanna Szpor, Janusz Ryś, Łukasz Szylberg, Jan P. Dumanski.

**Software:** Jakub Szymanowski.

**Supervision:** Natalia Filipowicz, Arkadiusz Piotrowski, Jan P. Dumanski.

**Visualization:** Natalia Filipowicz, Monika Horbacz, Jan P. Dumanski.

**Writing – original draft:** Natalia Filipowicz, Monika Horbacz, Jakub Szymanowski, Jan P. Dumanski.

**Writing – review & editing:** Natalia Filipowicz, Kinga Drężek, Monika Horbacz, Agata Wojdak, Jakub Szymanowski, Edyta Rychlicka-Buniowska, Ulana Juhas, Katarzyna Duzowska, Tomasz Nowikiewicz, Wiktoria Stańkowska, Katarzyna Chojnowska, Maria Andreou, Urszula Ławrynowicz, Magdalena Wójcik, Hanna Davies, Ewa Śrutek, Michał Bieńkowski, Katarzyna Milian-Ciesielska, Marek Zdrenka, Aleksandra Ambicka, Marcin Przewoźnik, Agnieszka Harazin-Lechowska, Agnieszka Adamczyk, Jacek Kowalski, Dariusz Bała, Dorian Wiśniewski, Karol Tkaczyński, Krzysztof Kamecki, Marta Drzewiecka, Paweł Wroński, Jerzy Siekiera, Izabela Ratnicka, Jerzy Jankau, Karol Wierzba, Jarosław Skokowski, Karol Połom, Mikołaj Przydacz, Łukasz Bełch, Piotr Chłosta, Marcin Matuszewski, Krzysztof Okoń, Olga Rostkowska, Andrzej Hellmann, Karol Sasim, Piotr Remiszewski, Marek Sierżęga, Stanisław Hać, Jarosław Kobiela, Łukasz Kaska, Michał Jankowski, Diana Hodorowicz-Zaniewska, Janusz Jaszczyński, Wojciech Zegarski, Wojciech Makarewicz,

Rafał Pęksa, Joanna Szpor, Janusz Ryś, Łukasz Szylberg, Arkadiusz Piotrowski, Jan P. Dumanski.

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
