## [Decision Letter · Decision Letter 0]

3 Dec 2021

PONE-D-21-28020Comprehensive cancer-oriented biobanking resource of human samples for studies of post-zygotic genetic variation involved in cancer predispositionPLOS ONE

Dear Dr. Filipowicz,

Thank you for submitting your manuscript to PLOS ONE. After careful consideration, we feel that it has merit but does not fully meet PLOS ONE’s publication criteria as it currently stands. Therefore, we invite you to submit a revised version of the manuscript that addresses the points raised during the review process.

We look forward to receiving your revised manuscript.

Kind regards,

Isaac Yi Kim, MD, PhD, MBA

Academic Editor

PLOS ONE

Journal Requirements:

2. For descriptions of databases/biobanks, please provide details about how the data were curated, as well as plans for long-term database/biobank maintenance, growth, and stability. Please provide a direct link to the database/biobank hosting site from within the paper. See our submission guidelines for further details  at http://journals.plos.org/plosone/s/submission-guidelines#loc-methods-software-databases-and-tools

"This study was sponsored by the Foundation for Polish Science (FNP) under the International Research Agendas Program (grant number MAB/2018/6) to JPD and AP, co-financed by the European Union under the European Regional Development Fund. Our collection also received financing via the “Excellence Initiative - Research University'' program from Medical University of Gdansk. This project obtained further partial funding from The Swedish Cancer Society and Swedish Medical Research Council to JPD"

"This study was sponsored by the Foundation for Polish Science (FNP) under the International Research Agendas Program to J.P.D. and A.P., co-financed by the European Union under the European Regional Development Fund. Our biobank also received financing via the “Excellence Initiative - Research University'' program from Medical University of Gdansk. This project obtained further partial funding from The Swedish Cancer Society and Swedish Medical Research Council to J.P.D."

5. We note that Figure 1 in your submission contain copyrighted images. All PLOS content is published under the Creative Commons Attribution License (CC BY 4.0), which means that the manuscript, images, and Supporting Information files will be freely available online, and any third party is permitted to access, download, copy, distribute, and use these materials in any way, even commercially, with proper attribution. For more information, see our copyright guidelines: http://journals.plos.org/plosone/s/licenses-and-copyright.

Reviewers' comments:

Reviewer's Responses to Questions

**Comments to the Author**

1. Is the manuscript technically sound, and do the data support the conclusions?

Reviewer #1: Yes

Reviewer #2: Yes

2. Has the statistical analysis been performed appropriately and rigorously? 

Reviewer #1: Yes

Reviewer #2: N/A

3. Have the authors made all data underlying the findings in their manuscript fully available?

Reviewer #1: Yes

Reviewer #2: Yes

4. Is the manuscript presented in an intelligible fashion and written in standard English?

Reviewer #1: Yes

Reviewer #2: Yes

5. Review Comments to the Author

Reviewer #1: The manuscript describes large scale biobanking that can be utilized for genomic analysis for cancer predisposition.

- As the authors mentioned, prostatic adenocarcinoma is usually multifocal and difficult to identify macroscopically. How are the PT and UM tissues verified that they in fact contain tumor and non-tumor tissues, respectively?

- Similarly, post-treatment tumors are difficult to identify macroscopically. How are the tissues verified?

- The protocol described is dependent on the presumptive diagnoses and macroscopic analysis for tissue collection. At times, the final pathology reports will not match the presumptive diagnoses. The procedure to address this issue needs to be explained.

Reviewer #2: Reviewer Comments

The biobanking approach by the authors in not an innovative idea or approach. However, the authors did an excellent and very thorough job in demonstrating a useful model of specimen coordination and tissue collection from multiple satellite sites. Below are questions and comments that should be addressed in the main text of the manuscript, which will help clarify missing details of the biobanking processes.

- Please indicate how specific regions of tumor harvest were determined? For example, was MRI used to confirm lesion or no lesion regions for the prostate cancer patients? What was the percentage of total number of tumors actually collected compared to the total number of tumors attempted to be collected (for each tumor type)?

- What considerations were taken in collecting samples from tumors with high heterogeneity? For each fragment of collected solid tissue, it was mentioned that a control tissue taken for H&E? How were “uninvolved margin” regions confirmed?

- For tumors that were used for multiple techniques such as H&E, OCT, etc., was the tumor split from the original sample or was adjacent tumor regions taken? If the later, how do you address potentially heterogeneity concerns during collection?

- How are tissue handled that are not 100% cancer? For example, what were done with tumor cores that contained a percentage of non-cancer tissue or tumor cores that had non-continuous cancer? How did this impact the freezing/storage of the tissue pieces? Were the non-cancer regions dissected from the cancer regions before freezing/storage?

- What specific techniques (such as frozen sections) were used by the pathologists in determining non-tumor regions and cancer regions? Were frozen sections taken at the time of surgery or during tissue processing to confirm the cancer and non-cancer regions for every tissue sample collected including tumor, UM regions, and matched controls?

- For breast cancer tissue collection, it was mentioned that a subset of tissues were collected directly from the surgical operations. Was the pathology of these tissue pieces also confirmed by frozen sections before making organoid and/or primary cell lines?

- What procedures/protocols were used to determine if local mets would be collected? Was the collection of mets pre-planned? How was biobanking and patient care both satisfied during met collection? In general, how were the mets collected/processed (i.e., FFPE, OCT, etc.)?

- For bladder cancer, TURBT procedures can be used for clinical diagnosis/staging in order to determine effective treatment plans. Did you encounter any embargoing of the tissue from TURBT procedures especially for heterogeneous areas of the bladder? Were there any issues with tissue collection for the biorepository that was canceled due to the need for clinical diagnosis/staging? If not, please explain how this was avoided?

- Were the pathologists used for biobanking all sub-specialized? For examples, was a GU pathologist used for bladder and prostate tissue collection? Why was it or why not was it necessary to have such sub-specialists in place?

- It was mentioned that “…collection from 1711 donors sums up to about 2800 working hours or about 70 working weeks with full time effort.” During the tissue collection pipeline, how much extra time was added to the clinical pipelines of surgery and tissue processing? Can you breakdown the 2800 working hours by task? For example, how many pathologists’ hours were needed?

- “The collected samples can be made available to other research groups.” Who gets to use the tissues? Is there a system in place to share the collected tissue? How will tissue usage be prioritized?

6. PLOS authors have the option to publish the peer review history of their article (what does this mean?). If published, this will include your full peer review and any attached files.

Reviewer #1: No

Reviewer #2: No

---

## [Author Response · Author response to Decision Letter 0]

16 Jan 2022

Re: revision of the manuscript PONE-D-21-28020 “Comprehensive cancer-oriented biobanking resource of human samples for studies of post-zygotic genetic variation involved in cancer predisposition”

General comment regarding answers to the below questions/comments from Reviewer #1 and Reviewer #2

There is a large overlap between the issues raised and we reply by cross-referring between our answers to both of the reviewers, to avoid unnecessary repetitions of the text. Therefore, we believe that all our answers should be shown to both reviewers. 

Reviewers' comments and our answers:

Reviewer #1: The manuscript describes large scale biobanking that can be utilized for genomic analysis for cancer predisposition. - 

1. As the authors mentioned, prostatic adenocarcinoma is usually multifocal and difficult to identify macroscopically. How are the PT and UM tissues verified that they in fact contain tumor and non-tumor tissues, respectively?

2. Similarly, post-treatment tumors are difficult to identify macroscopically. How are the tissues verified?

3. The protocol described is dependent on the presumptive diagnoses and macroscopic analysis for tissue collection. At times, the final pathology reports will not match the presumptive diagnoses. The procedure to address this issue needs to be explained.

Reviewer #1: Answer to questions 1-3: 

These three questions are closely related and their primary relevance is prostatic adenocarcinoma, because of recurrent problems to macroscopically define the location of primary tumor. The second major diagnosis where these questions are also quite relevant is identification of multifocal breast cancer. In our past experience, UMs taken from areas of macroscopically normal breast tissue (close, or occasionally at considerable distances from the primary tumor; see PMIDs 26430163 and 26219265) may sometimes contain tumor cells. Therefore, it is obviously important to characterize histologically each tissue fragment (presumed macroscopically normal or suspected to contain tumor tissue) that is biobanked, which we did (and continue collecting in this way, as the biobanking project is not yet finished). 

The tissues were collected according to the well-defined protocol to address the above issue, i.e., after macro-sectioning of the resected organ, small tissue fragments were selected and excised for biobanking. Subsequently, each fragment was cut in half: one portion was placed into a cryovial and fresh-frozen at -80℃, while the other one was fixed in formalin, embedded in paraffin and underwent standard processing, sectioning and H&E staining (FFPE). The latter FFPE tissue sectioning was done along the cut surface closest to the fresh-frozen biobanked piece of tissue, so that the FFPE section is as much as possible representative for the tissue in the frozen specimen. 

For instance, for prostate specimens, a total of twelve tissue fragments were biobanked, each followed by the pathological report on the actual tumor content in the matching sample. Therefore, despite the degree of uncertainty/discrepancy in the macroscopic assessment in some situations (i.e. for prostatic adenocarcinoma, multifocal breast cancer, pancreatic adenocarcinoma with coexistent chronic pancreatitis, tumors after neoadjuvant therapy), every single biobanked tissue fragment has its matching FFPE tissue that undergoes pathological verification of the actual tumor content. The results of this assessment is every time included in the routine patient histopathological report. 

The above questions are similar to some of questions/comments from reviewer #2. Because of these concerns, we have modified our manuscript to emphasize this issue (please see page 8 and 9) . Furthermore, it is important to mention that each time, the biobanked material is selected for further molecular analysis(es), the presence/absence of tumor/non-tumor tissue is being verified using the pathological reports from the participating hospitals. If necessary, an additional investigation of FFPE slides for both primary tumors and UMs that are archived, can be undertaken. 

Reviewer #2: Reviewer Comments

The biobanking approach by the authors in not an innovative idea or approach. However, the authors did an excellent and very thorough job in demonstrating a useful model of specimen coordination and tissue collection from multiple satellite sites. Below are questions and comments that should be addressed in the main text of the manuscript, which will help clarify missing details of the biobanking processes. 

1. Please indicate how specific regions of tumor harvest were determined? For example, was MRI used to confirm lesion or no lesion regions for the prostate cancer patients? 

For most included patients and cancer types, the tumor is clearly identifiable macroscopically. However, as mentioned above for prostate cancer, the macroscopic assessment is sometimes unreliable. Therefore, pre-specified regions of the gland (central and peripheral areas of upper/central/lower part of each lobe, similarly to the mapping biopsy procedure) were harvested. Subsequently, all matching specimens underwent pathologic examination to verify the actual tumor content (see above answers to reviewer #1). 

Whenever imaging analysis was performed prior to surgery, MRI and/or CT results were accessible for pathologists prior to collection tumor/non-tumor tissue from the prostate gland. We do not currently have data on how often MRI and/or CT results were available to the pathologists and how efficient these imaging data actually were in guiding pathologists during the macro-sectioning of the prostate. However, we confirm that the pathologists were using the MRI and/or CT results during macro-dissection of the prostate. This issue is actually an interesting future research question that could be addressed when we finish our collection of prostate cancers. We could then evaluate (using a much larger sample size) the effectiveness of targeting the primary tumor by the pathologist, when dissecting prostate. 

Generally speaking, our protocol for prostate cancer sample collection was designed to cover most of the possible tumor locations, in order to sample both primary tumor and normal prostate tissue. Based on the presumable location of the tumor, we selected four likely tumor specimens (named PT1-4; see Figure 2C in the manuscript). Samples with presumably no lesions in the slice towards apex (LUM1-4) and towards base (UUM1-4) were analogically excised. All the specimens were further confirmed by post-operative histopathological examination. See also below answer to question 2. 

2. What was the percentage of total number of tumors actually collected compared to the total number of tumors attempted to be collected (for each tumor type)? 

Thank you for this comment, as this is an important aspect of our project. We presume that this question refers to the statistics of how often the presence of tumor tissue was verified by histopathological analysis of FFPE-sections in the samples that were macroscopically presumed to represent a tumor sample. 

In response to this question, we performed such comparison for prostate cancer and colorectal cancer. As suspected, prostate cancer represents a challenge due to the lack of clear macroscopic demarcation of the tumor. The calculation on the cohort of 100 prostate cancer patients showed that out of 400 specimens collected presumably as tumorous tissue, 263 (65.7%) showed the presence of tumor cells. For the similar sample size of 100 colorectal cancer patients, 205 out of 205 (100%) presumed tumor samples were confirmed as containing tumorous tissue. It is again important to mention (see also below question 3) that each fragment selected for further molecular examination is verified with the histopathological report. We added the text summarizing the above comparison in the results section of the paper, on page 18. 

We do not yet have similar statistics for other diagnoses. 

3. What considerations were taken in collecting samples from tumors with high heterogeneity? For each fragment of collected solid tissue, it was mentioned that a control tissue taken for H&E? How were “uninvolved margin” regions confirmed? 

Please see answers above. Each collected specimen has a matching FFPE fragment that undergoes the histological verification. This allows us to asses very well the heterogeneity of different pieces of the primary tumor and metastases. 

4. For tumors that were used for multiple techniques such as H&E, OCT, etc., was the tumor split from the original sample or was adjacent tumor regions taken? If the later, how do you address potentially heterogeneity concerns during collection?

Please see above the answer to reviewer #1. Each collected sample was split in half and the splitting surface underwent histological assessment of the actual tissue content (i.e. each resected tissue fragment had two halves: biobanked fragment and reference one for FFPE). This approach provides only a general representation of the actual content of the biobanked material, leaving some uncertainty margin, as tissue areas 1 mm apart may differ. It is, however, the best way that it can be done. We cannot be 100% certain that the tissue used for DNA/RNA/protein extraction is identical to what is present on the histological slide.

5. How are tissue handled that are not 100% cancer? For example, what were done with tumor cores that contained a percentage of non-cancer tissue or tumor cores that had non-continuous cancer? How did this impact the freezing/storage of the tissue pieces? Were the non-cancer regions dissected from the cancer regions before freezing/storage? 

It is safe to say that none of the tumor tissue specimens is containing 100% cancer, as cancer cells are always intermingled with a quite variable proportion of immune cells, fibroblasts, vessels etc. The histological verification of the tissue content was performed post-factum and there were no attempts to remove the non-neoplastic tissues. However, our samples can be used for future micro-dissection projects, if motivated by findings using bulk of the cells in a sample, to verify these results.

6. What specific techniques (such as frozen sections) were used by the pathologists in determining non-tumor regions and cancer regions? Were frozen sections taken at the time of surgery or during tissue processing to confirm the cancer and non-cancer regions for every tissue sample collected including tumor, UM regions, and matched controls?

We did not perform intraoperative pathological assessment (frozen sections) for the patients that were biobanked in our project, since this process is much more time consuming. We were aiming at shortening time between organ resection and snap freezing of the fragments (up to 2h), which is of importance for the quality of genetic material got later from the collected tissue fragments (especially RNA integrity). Moreover, intra-operative sampling decreases the amount of tumor tissue that is available for later diagnostic procedure using FFPE and material that can be biobanked. Using the approach of two halves of tissue fragments - biobanked and reference for FFPE (see above point 4), we could confidently confirm the content of biobanked material. The drawback of this approach is that the tissue fragments selected for harvesting based on the macroscopic assessment had to be verified after the surgery. Thus, as we mentioned above, each time, the frozen material is selected for further molecular analysis(es), the presence/absence of tumor/non-tumor tissue is being verified using the pathological reports from the participating hospitals.

7. For breast cancer tissue collection, it was mentioned that a subset of tissues were collected directly from the surgical operations. Was the pathology of these tissue pieces also confirmed by frozen sections before making organoid and/or primary cell lines? 

The procedure for UM (uninvolved margin) and S (skin) collection for primary cell cultures was the only exception, where the fragments of tissue were collected directly at the operating theatre due to the necessity to keep the strictly sterile conditions. The morphology for those fragments were not confirmed by microscopical assessment. However, the UM fragment for patients undergoing BCT was collected from the most distant possible location of the breast, in comparison to the location of primary tumor. For mastectomy patients, UM and S fragments were always collected from the opposite quadrant of the breast. 

8. What procedures/protocols were used to determine if local mets would be collected? Was the collection of mets pre-planned? How was biobanking and patient care both satisfied during met collection? In general, how were the mets collected/processed (i.e., FFPE, OCT, etc.)? 

The collection of local metastases was optional, in case of breast and colorectal carcinoma patients. To ensure proper diagnosis and patient care, nodal metastases were only collected if they were clearly identifiable and large enough on gross examination. The metastases were processed in the same way as the other specimens. We added respective information about that on page 8 of the manuscript.

9. For bladder cancer, TURBT procedures can be used for clinical diagnosis/staging in order to determine effective treatment plans. Did you encounter any embargoing of the tissue from TURBT procedures especially for heterogeneous areas of the bladder? Were there any issues with tissue collection for the biorepository that was canceled due to the need for clinical diagnosis/staging? If not, please explain how this was avoided? 

Collection of samples from TURBT procedures is the most challenging protocol considering the limited amount of tissue that is collected. The decision to enroll the patient into the biobanking project is taken by the surgeon and he/she takes into account several variables; e.g. size of the tumor and heterogeneity of the areas from which both tumor and a fragment(s) of normal tissue can be taken. However, the final decision to collect (or not) the fragments for biobanking is taken by the pathologist and this decision depends on the amount of tumor tissue that is available to safeguard the proper diagnostic procedure of the patient. The priority #1 is always the patient diagnosis. 

We have a histopathological control for the normal tissue also in the case of this diagnosis (after TURBT procedure). However, it should be mentioned that the amount of normal tissue that is available strongly limits the number and type of molecular analyses that can be performed. As it is clearly visible in Figure 3D of the manuscript, the number of samples taken during TURBT procedure is usually three: one PT and two UMs (close and distant UM samples PT), with the minimal mandatory set of two samples (one PT and one UM) than can be included in our biobanking project. 

In conclusion, there is no risk that the patient would suffer from problems regarding diagnosis, including the proper staging of the disease. If tumor sample is very small according to the experienced pathologist, we do not biobank material for such patient. If the patient is selected to biobank the tissue, the tumor sample as well as the fragment of normal tissue is divided into two “twin fragments”, one for FFPE and one for the biobank. 

10. Were the pathologists used for biobanking all sub-specialized? For examples, was a GU pathologist used for bladder and prostate tissue collection? Why was it or why not was it necessary to have such sub-specialists in place? 

In the larger cancer centers, where pathologists are diagnosing a very large number of patients, there usually is a degree of division of work involving specialization of pathologists for different diagnoses. This is valid only for physicians who already are specialist consultants in Pathology. 

For the purpose of our biobanking project, there was no need to involve sub-specialization, as the biobanking protocol was fairly simple, although time consuming for some diagnoses. In each participating hospital, there was (is) a team of pathologists, who perform gross examination of the resected organs, biobanked samples and later were involved in the assessment of the tumor content. This team is processing all diagnoses for our project that are coming from any of the five collaborating hospitals. All the pathologists involved in our project prepared material according to detailed collection protocols and after training. Secondly, the workload necessary for this endeavor required the participation of the whole (or vast majority of) pathology teams (in smaller hospitals the whole team may be limited to two physicians) as the specimens sometimes arrived irregularly and had to be processed directly upon arrival. Strict sub-specialization is rarely practiced in Poland and almost only in hospitals dedicated to a narrow group of diseases, e.g., Institute of Tuberculosis and Lung Diseases or in reference Breast Cancer Units. Finally, sub-specialization is necessary for rare diagnoses or uncharacteristic entities and our study included common tumors, most of which had already been diagnosed in core needle biopsy or endoscopy specimens. 

Occasional discrepancies and problems between the final diagnosis for the resected organ and the biopsy material may occur, but the assessment of the samples reflecting the biobanked tissues was always performed in parallel to the final pathologic diagnosis by the same pathologist, while any uncertainties were consulted with other team members. 

11. It was mentioned that “… collection from 1711 donors sums up to about 2800 working hours or about 70 working weeks with full time effort.” During the tissue collection pipeline, how much extra time was added to the clinical pipelines of surgery and tissue processing? Can you breakdown the 2800 working hours by task? For example, how many pathologists’ hours were needed?

The below table is the summary of the requested details. 

This information is a rough estimation of the extra time spent by the personnel on our biobanking project; that means the time added on top of what would be the time necessary to process a patient without biobanking. The presented numbers reflect the mean estimates for each task provided by the four largest partner hospitals: 

Procedure Blood preparation Sample registration in the biobanking system Sample preparation, slide preparation and assessment in the pathology department Total

Breast-BCT 10 min 11 min 50 min 71 min

Breast-Mastectomy 10 min 14 min 81 min 105 min

Prostatectomy 10 min 16 min 128 min 154 min

Colectomy 10 min 15 min 65 min 90 min

Cystectomy 10 min 15 min 77 min 92 min

Hepatectomy 10 min 15 min 35 min 60 min

Pancreatectomy 10 min 15 min 85 min 110 min

The clinical pipeline of the surgery was longer only in two cases of our biobanking activity: while collecting the UM fragment from the glandular tissue of the breast for cell cultures directly at operating theatre (up to 5 min.) and during TURBT procedure (up to 5 min.).

We believe that this detailed calculation of time is not the most important aspect of the manuscript. Our aim was merely to provide an estimation of the necessary workload, that could be used by other investigators for planning purposes. 

12. The collected samples can be made available to other research groups. ”Who gets to use the tissues? Is there a system in place to share the collected tissue? How will tissue usage be prioritized?” 

Thank you for these questions, which point to an insufficient description of this aspect. The collected biological material and clinical details can be made available for other investigators after a request to both corresponding authors. The letter should outline the aim, number/type of requested samples and methodology of the proposed collaborative project. A sentence similar to the above one, has been added to the main text of the manuscript. 

We obtained the research ethical approval, which is according to “the model of broad consent”, for the biobanking project outlined in our application for approval. This allows us to share, on the collaborative basis, the collected biological samples (in many formats; e.g. tissues, serum, blood and processed biological material such as DNA, RNA, etc.; all as pseudo-anonymized specimens) with investigators in Poland and outside for the purpose of specified collaboration project. Clinical details are available only to the authorized personnel of our unit and can be made available to outside investigators upon request. 

The priority is determined by the availability of material, considering projects that are already ongoing, and avoiding competition for the same research questions. 

Natalia Filipowicz

Jan Dumanski

---

## [Editor Report · Decision Letter 1]

15 Mar 2022

Comprehensive cancer-oriented biobanking resource of human samples for studies of post-zygotic genetic variation involved in cancer predisposition

PONE-D-21-28020R1

Dear Dr. Filipowicz,

We’re pleased to inform you that your manuscript has been judged scientifically suitable for publication and will be formally accepted for publication once it meets all outstanding technical requirements.

Kind regards,

Isaac Yi Kim, MD, PhD, MBA

Academic Editor

PLOS ONE
---

## [Editor Report · Acceptance letter]

30 Mar 2022

PONE-D-21-28020R1 

Comprehensive cancer-oriented biobanking resource of human samples for studies of post-zygotic genetic variation involved in cancer predisposition 

Dear Dr. Filipowicz:

I'm pleased to inform you that your manuscript has been deemed suitable for publication in PLOS ONE. Congratulations! Your manuscript is now with our production department. 

Kind regards, 

on behalf of

Dr. Isaac Yi Kim 

Academic Editor

PLOS ONE